# What Does PET Imaging Bring to Neuro-Oncology in 2022? A Review

**DOI:** 10.3390/cancers14040879

**Published:** 2022-02-10

**Authors:** Jules Tianyu Zhang-Yin, Antoine Girard, Marc Bertaux

**Affiliations:** 1Department of Nuclear Medicine, Clinique Sud Luxembourg, 486762 Arlon, Belgium; 2Department of Nuclear Medicine, Centre Eugène Marquis, Université Rennes 1, 35000 Rennes, France; a.girard@rennes.unicancer.fr; 3Department of Nuclear Medicine, Foch Hospital, 92150 Suresnes, France; m.bertaux@hopital-foch.com

**Keywords:** [^18^F]FDG, [^18^F]FDOPA, [^18^F]FET, [^68^Ga]Ga-DOTA-SSTR, glioma, brain metastases, meningioma, PCNSL

## Abstract

**Simple Summary:**

Positron emission tomography (PET) imaging is increasingly used to supplement MRI in the management of patient with brain tumors. In this article, we provide a review of the current place and perspectives of PET imaging for the diagnosis and follow-up of from primary brain tumors such as gliomas, meningiomas and central nervous system lymphomas, as well as brain metastases. Different PET radiotracers targeting different biological processes are used to accurately depict these brain tumors and provide unique metabolic and biologic information. Radiolabeled amino acids such as [^18^F]FDOPA or [^18^F]FET are used for imaging of gliomas while both [^18^F]FDG and amino acids can be used for brain metastases. Meningiomas can be seen with a high contrast using radiolabeled ligands of somatostatin receptors, which they usually carry. Unconventional tracers that allow the study of other biological processes such as cell proliferation, hypoxia, or neo-angiogenesis are currently being studied for brain tumors imaging.

**Abstract:**

PET imaging is being increasingly used to supplement MRI in the clinical management of brain tumors. The main radiotracers implemented in clinical practice include [^18^F]FDG, radiolabeled amino acids ([^11^C]MET, [^18^F]FDOPA, [^18^F]FET) and [^68^Ga]Ga-DOTA-SSTR, targeting glucose metabolism, L-amino-acid transport and somatostatin receptors expression, respectively. This review aims at addressing the current place and perspectives of brain PET imaging for patients who suffer from primary or secondary brain tumors, at diagnosis and during follow-up. A special focus is given to the following: radiolabeled amino acids PET imaging for tumor characterization and follow-up in gliomas; the role of amino acid PET and [^18^F]FDG PET for detecting brain metastases recurrence; [^68^Ga]Ga-DOTA-SSTR PET for guiding treatment in meningioma and particularly before targeted radiotherapy.

## 1. Introduction

Multiparametric magnetic resonance imaging (mpMRI) is the reference method for the diagnosis and follow-up of primary and secondary brain tumors. Nevertheless, its ability to distinguish between viable neoplastic tissue and tumor-free areas is limited in many cases [1,2,3]. This is especially true in the post-radiation therapy setting where treatment-related changes (TRC) can mimic tumor progression. Over the past few decades, various positron emission tomography (PET) tracers have emerged in the field of neuro-oncology. These radiolabeled molecules can be used as a complementary tool to overcome some of mpMRI’s limitations, as follows: to help discriminate TRC changes from tumor progression or relapse, to delineate the tumor extent, to highlight non-enhancing tumors, and to monitor treatment response and to predict the patients’ outcome [4,5].

This narrative review aims at addressing the current place and perspectives of brain PET imaging for patients who suffer from primary or secondary brain tumors, at diagnosis and during follow-up. The main clinical applications are as follows: the delineation of tumor extent for treatment planning, assessment of treatment response in gliomas, differential diagnosis and detection of meningioma, distinction of post-therapeutic reactive changes after radiotherapy from progression or recurrence in glioma and brain metastases and differential diagnosis in primary central nervous system lymphoma (PCNSL). The PET tracers involved are ^18^F-2-fluoro-2-deoxy-d-glucose ([^18^F]FDG), radiolabeled amino acids ([^11^C]methionine ([^11^C]MET), 3,4-dihydroxy6-^18^F-fluoro-l-phenylalanine ([^18^F]FDOPA), O-(2-^18^F fluoroethyl)-l-tyrosine ([^18^F]FET)) and [^68^Ga]Ga-DOTA-somatostatin receptors (SSTR). A brief look at the emerging aspect of theragnostic, especially for [^177^Lu]Lu-DOTA-SSTR in the treatment of meningioma, is also included.

## 2. Tracers

In general oncology, [^18^F]FDG is a well-established and the most widely used tracer for PET imaging [6]. An increased FDG uptake corresponds to increased glucose metabolism. It is commonly seen in malignancy because of glucose transporter 1 (GLUT1) over-expression and increased hexokinase phosphorylation. Nonetheless, the physiological high FDG uptake in the normal brain, due to neurons’ activities, limits the lesion-to-background contrast for brain tumors [5]. Nevertheless, [^18^F]FDG remains useful for intensely hypermetabolic brain lesions such as PCNSL, glioblastoma and some metastases.

Amino acid PET tracers have the advantage over FDG in that they do not accumulate too much in the normal brain. [^11^C]MET has been the first developed tracer in this field, but its short half-life (20 min) limits its availability to centers equipped with in-house cyclotron.

Several amino acid PET tracers radiolabeled with ^18^F (half-life:110 min) have been developed since then, including [^18^F]FDOPA and [^18^F]FET, which are the most often used. Their uptake in brain tumors relies on the overexpression of large amino acid transporters of the L-type (LAT) [7].

Another PET tracer family of interest is ^68^Ga-labeled SSTR ligands, since somatostatin receptors are overexpressed in meningioma, with DOTA-D-Phe1-Tyr3-octreotate (DOTATATE) and DOTA-Tyr3-octreotide (DOTATOC) being the most used in clinical routine. They usually require an in-house gallium generator as its half-life is quite short (68 min) but can also be shipped over short distances.

Other PET tracers designed to study various biological aspect of brain tumors have been developed or are currently being developed, but they are still rarely used in clinical routine. It is always important to distinguish radiotracers regarding their ability to cross the healthy blood–brain barrier. [^18^F]FDG and radiolabeled amino acids ([^11^C]MET, [^18^F]FDOPA, [^18^F]FET) do cross the healthy blood–brain barrier, thus allowing researchers to image infiltrative tumors that do not disrupt it [8,9].

## 3. Glioma

Amino acid PET tracers have relatively high sensitivity for gliomas as the LAT system transporters is overexpressed in most of them [10], with a good tumor-to-background contrast in the brain [11]. Thus, their use has been recommended by the Response Assessment Neuro-Oncology group (RANO) for the assessment of glioma in many situations [5].

### 3.1. Diagnostic and Characterization

[^11^C]MET is chemically equivalent to natural methionine it is incorporated into proteins and could therefore accumulate over time within tumors [12]. Nevertheless, ^11^C short physical half-life limits its availability and does not allow for delayed imaging. [^11^C]MET has good performance to diagnose brain tumors as highlighted in a meta-analysis including more than 400 patients by Zhao et al. [13]. In their study, [^11^C]MET had a high pooled sensitivity and specificity of 91% and 86% for neoplastic tissue, whereas those of ^18^F-FDG were only moderate with sensitivity and specificity of 71% and 77%, respectively.

Nowadays, the emergence of ^18^F radiolabeled amino acid tracers such as [^18^F]FDOPA and [^18^F]FET enabled a wide use of amino acid PET in clinical practice in many centers. In general, they have comparable diagnostic value compared to [^11^C]MET [14,15]. 

[^18^F]FDOPA PET has a good accuracy for the diagnosis of primary brain tumors, with a sensitivity of 96% and a specificity of 86% [16]. It is more specific than [^18^F]FDG [16] and performs as well as ^11^C-MET [17]. The visual and semi-quantitative analyses can both be used. In the visual analysis, the positivity can be defined by a lesion’s uptake greater than or equal to striatum uptake. In the semi-quantitative analysis, the only pathology-controlled thresholds to detect brain tumors is a ratio of 1.0 for tumor-to-striatum and 1.3 for tumor-to-brain [16,18] (Figure 1). It can also be used to differentiate high- and low-grade gliomas as the uptake is significantly higher in high-grade gliomas [19]. A recent meta-analysis found a pooled sensitivity of 0.88 and a pooled specificity and 0.73 for glioma grading [20], making it a valuable clinical tool [21]. The dynamic analysis also seems to be an interesting tool to integrate [22,23,24]. At diagnosis [^18^F]FDOPA uptake also has an independent prognostic value [25]. 

In the same line, [^18^F]FET is another very performant tracer as a meta-analysis of 13 [^18^F]FET PET studies, including more than 450 patients, showed pooled sensitivity and specificity both around 80% for the diagnosis of primary brain tumors [26]. More recently, a second meta-analysis, including 119 patients, reported pooled sensitivity and specificity of 0.94 and 0.88, respectively [27]. A mean tumor-to-background uptake threshold ratio of at least 1.6 and a maximum TBR of at least 2.1 seems to be the best cutoff. Nevertheless, the same authors, in another previous study, have found some contradictory results, especially a relatively low specificity of 62% for a good sensitivity of 84% [26]. The threshold is 2.1 for tumor-to-brain (TBR) max et 1.7 for TBR mean to distinguish glioma versus non-glioma [26]. It allows the biopsy guidance [9] and differentiate efficiently high- and low-grade gliomas, especially thanks to parameters derived from dynamic acquisition [28].

It is particularly important to rule out false positivity, as increased amino acid tracer uptake may also occur in nonneoplastic lesions or inflammatory processes such for [^18^F]FET and for [^18^F]FDOPA [29,30,31], but uptake intensity is generally low or moderate, below the intensity of high-grade gliomas’ uptake.

Regarding false negativity, it is mainly related to isocitrate dehydrogenase (IDH)-mutated gliomas. Up to 30% of WHO grade II IDH-mutated gliomas do not show significant amino acid uptake; thus, negative amino acid PET is not sufficient to rule-out low-grade glioma [7,32,33]. 

### 3.2. Defining Tumor Extent

Delineating glioma extent is important for further diagnostic and therapeutic management, such as biopsy, resection, or radiotherapy planning (Figure 1). To do so [^18^F]FDOPA and [^18^F]FET can complement mpMRI. Indeed, conventional mpMRI is particularly limited in its ability to identify non-enhancing glioma subregions [1], whereas [^18^F]FDOPA PET could discriminate glioma from benign brain lesions such as dysembryoplastic neuroepithelial tumor, and high- from low-grade gliomas, with no contrast enhancement on mpMRI [19]. In a recent biopsy-validated study, molecular information obtained from [^18^F]FET would reveal a more accurate glioma extent, which is critical for individualized treatment planning [34].

As there is a significant difference in evaluating tumor volume between amino acid PET and mpMRI, it suggests that the latter could substantially underestimate the metabolically active tumor volume [35,36]. Another recent evidence-based article suggested that amino acid PET could improve the delineation of high-grade gliomas compared to standard mpMRI [37]. However, till today, only one study has demonstrated that the management based on amino acid PET-guided benefits a better patient outcome [38].

### 3.3. Defining Tumor Heterogeneity

The extent of tumor resection is one of the most important prognostic factors in gliomas. Nevertheless, often these tumors are highly heterogeneous with a possible coexistence of high- and low-grade subregions. Thus, presurgical identification of high-grade subregions extent is of major importance. In this very recent biopsy validation study, Girard et al. found that the addition of [^18^F]FDOPA PET to mpMRI enlarged the delineation volumes and enhanced overall accuracy for detection of high-grade subregions. Thus, combining [^18^F]FDOPA PET with advanced mpMRI may improve treatment planning in newly diagnosed gliomas [39]. In another clinical trial, better defining tumor heterogeneity seems to have a high impact on radiation therapy. Indeed, [^18^F]FDOPA PET-guided dose-escalated radiation therapy significantly improved the overall survival in the subgroup of O^6^-methylguanine-DNA methyltransferase (MGMT) methylated glioblastoma patients and the progression-free survival in MGMT unmethylated glioblastoma patients [38].

### 3.4. Monitoring Therapy

In gliomas, frequently used systemic treatment options are alkylating chemotherapy and antiangiogenic therapy, associated with radiotherapy (protocol STUPP). The mpMRI method remains the standard tool for assessment of response but it often lacks specificity [3].

[^18^F]FET can provide a reliable response assessment after chemotherapy (temozolomide and nitrosourea-based) in patients with high-grade glioma at recurrence [39]. It also can predict patients’ outcomes as metabolic responders have better survival [40,41].

An early [^18^F]FET uptake change, seen 6 weeks after chemoradiotherapy using temozolomide, could already have prognostic information. A decrease in metabolic activity of more than 10% can be a cutoff for predicting better patients’ survival [42].

Contrary to [^18^F]FET, there is less evidence for [^18^F]FDOPA. There are two studies on assessment after anti-angiogenic treatments, and they concluded that [^18^F]FDOPA PET could monitor response and identify responders after 2 weeks of treatment with bevacizumab [43,44].

### 3.5. Recurrence vs. Radionecrosis

Differentiating TRC from disease progression is of critical importance for patients’ management and prognosis and it can often be challenging [45]. In some countries such as France, suspected recurrent or progressive glioma after treatment is the most common indication for [^18^F]FDOPA PET in brain tumors. The mpMRI method has some limitations in the case of pseudo-progression in the first 12 weeks after treatment, as in the case of radio-necrosis after 12 weeks of treatment [45,46]. These TRC can appear like a new or increasing contrast enhancement that cannot be distinguished from tumor recurrence. 

The [^18^F]FDOPA PET has an accuracy from 82% to 96% to distinguish TRC vs. a progressive tumor [47,48]. The visual criteria, using the striatum contralateral to the tumor as the reference, or semi-quantitatively, with a tumor-to-striatum ratio of 1.4 for SUVmax and 1.2 for SUV mean, could both be effective [47] (Figure 2). 

The [^18^F]FET has a similar performance [44], whereas the [^11^C]MET encounters a slightly lower accuracy of approximately 75% [49] as MET has more affinity for inflammatory processes [50].

## 4. Metastases

Brain metastases (BM) are the most common malignant brain tumors, as they are part of the natural course of several types of cancer, especially breast and lung cancer, as well as melanomas. Contrast-enhanced mpMRI is the cornerstone of metastatic brain tumor evaluation. It has widespread availability and excellent spatial resolution, but its specificity can be low, especially in distinguishing TRC from progression, resulting in substantial diagnostic challenges [51]. Amino acid PET tracers can be useful because, as gliomas, metastatic brain tumors overexpress LAT, independently of the primitive tumor. However, [^18^F]FDG has also a role to play.

### 4.1. Diagnostic and Characterization

For the detection of BM, the [^18^F]FDG has poor accuracy, as highlighted in a recent meta-analysis including more than 900 lung cancer patients with brain metastases and comparing contrast-enhanced mpMRI to [^18^F]FDG PET. It was found that mpMRI has a substantially higher cumulative sensitivity (77%) than [^18^F]FDG PET (21%) [52]. 

Amino acid PET tracers are substantially more performant than [^18^F]FDG, especially for lesions more than 1 cm. Unterrainer et al. reported that [^18^F]FET were positive for approximately 90% BM, using a ratio ≥ 1.6 for tumor/brain [53]. For lesions smaller than 1 cm, the detection rate by mpMRI remains the best, nearly 100% [53]. So, mpMRI is the reference imaging modality for the detection of brain metastases.

There is limited evidence to support the use of PET to distinguish between BM and other brain tumors, especially high-grade glioma [52]. Some authors reported that [^18^F]FDG is generally lower in metastases than in PCNSL and amino acid PET tracers could identify aggressive tumor features and thus predict a worse prognosis [54,55]. 

### 4.2. Detecting Occult Primary Extracerebral Malignancy Revealed by Brain Metastases

In the case of newly discovered BM in patients with no history of cancer, primary lesion and other extracerebral metastases must be sought. Several studies investigating [^18^F]FDG PET revealed its good performance.

Roh et al. showed that the sensitivity of [^18^F]FDG PET (87.5%) was significantly higher than that of CT (43.7%) in the detection of the primary tumor in patients with BM [56]. The main sites of other extracerebral metastases are in lymph nodes, especially mediastinal, hilar and retroperitoneal ones [57,58], and the lung was the most frequent primary tumor in patients with brain metastases [58,59]. Moreover, [^18^F]FDG PET detected additional extracerebral metastatic sites in 42% to 63% of patients [57,58,59,60].

### 4.3. Recurrence vs. Radionecrosis

Among irradiated patients, brain radionecrosis is a common complication, mainly depending on irradiation technique [61], which occurs with an incidence up to 25% [62]. Most of the brain radionecroses are diagnosed during the year after the end of radiotherapy and in 80% within 3 years [63]. Differentiating brain metastases recurrence from radionecrosis can be challenging during mpMRI follow-up after stereotactic radiotherapy.

According to the RANO/PET working group, amino acids PET tracer should be preferred in this indication [51,64]. Reported diagnostic performance of amino acids PET is high and reproducible with sensitivity ranging between 74 and 90% and specificity between 75 and 100% [65,66,67,68,69,70,71,72]. 

Reported sensitivity of standard [^18^F]FDG ranges from 40% to 83% and specificity between 50% and 94%, respectively [63,73,74,75]. This limited diagnostic performance is mainly due to low tumor-to-brain on standard images. To overcome this issue, some authors performed additional delayed PET images 4 to 5 h after [^18^F]FDG injection. With such protocols, [^18^F]FDG PET reached sensitivity of 93–95% and specificity of 94–100% [76,77] (Figure 3). 

## 5. Meningioma

Meningioma is the most common non-glial primary brain tumor, which represents approximately 35% of all brain tumors. A high SSTR type 2 density is found in all meningioma [78]. [^68^Ga]Ga-DOTA-SSTR PET tracers are SST analogs with a high binding affinity to SSTR type 2, which make them an effective tool for imaging meningioma.

### 5.1. Diagnostic and Characterization

Detecting meningioma can be challenging using mpMRI, notably when it locates at the skull base or nearby the falx cerebri, often with extension to adjacent bone structure. [^68^Ga]Ga-DOTA-SSTR has high sensitivity in the detection of meningioma compared to contrast-enhanced mpMRI [79]. Except the pituitary gland, there is no physiological uptake in the brain for [^68^Ga]Ga-DOTA-SSTR, which provides a high tumor-to-background ratio. The specificity for meningioma is not perfect because [^68^Ga]Ga-DOTA-SSTR also show a moderate uptake in inflammatory lesions [80]. [^68^Ga]Ga-DOTA-SSTR PET can be particularly useful for differential diagnosis between meningioma and other kinds of tumors with low SSTR expression, such as schwannoma [81]. 

### 5.2. Defining Tumor Extent

[^68^Ga]Ga-DOTA-SSTR PET can delineate more precisely the tumor extent in various tumor locations than contrast-enhanced mpMRI, in a comparative study with histological confirmation [82]. This added value is more pronounced when the tumor is located in regions such as the skull base, orbit and cavernous sinus [83,84] or optic pathway [85]. This is particularly true in meningioma that were previously treated with surgery and/or radiotherapy to differentiate post-therapeutic changes from active meningioma.

Before radiotherapy, it is a valuable tool for improving the GTV (gross tumor volume) and the CTV (clinical target volume) definition and sparing the organs at risk [80].

This optimized target volume delineation is of great help for stereotaxic fractionated radiotherapy in grade I-III meningiomas [86].

In a head-to-head comparison study, [^68^Ga]Ga-DOTA-SSTR PET performed better than [^18^F]FET PET in GTV definition. Overall, 2 out of 21 false-negative patients were reported with [^18^F]FET PET, whereas the [^68^Ga]Ga-DOTA-SSTR PET had 100% sensitivity [87].

### 5.3. Assessment of Response to Radiotherapy

Only a few data are available in the literature. [^11^C]MET PET seems to be of interest in the early response assessment after high-energy proton therapy with an average uptake intensity reduction of 19.4% [88]. Ryttlefors et al. stated that a follow-up with [^11^C]MET PET may be a valuable adjunct to, but not a replacement for, standard radiological follow-up [89].

More recently, [^68^Ga]Ga-DOTA-DOTATATE PET has been proven to be a reliable tool in evaluating treatment responses to radiation therapy. The mean and the maximal total lesion activities decreased significantly with a median of 14.7% [range 8.5–97%] and 36% [range 15–105%], respectively, while the tumor volume based on mpMRI measurement did not change significantly, according to RECIST criteria. Thus, the [^68^Ga]Ga-DOTA-SSTR PET has incremental value for assessing the treatment response [90].

### 5.4. Diagnosis of Recurrence after Surgery

The recurrence rate is estimated between 20 and 40% within ten years, despite macroscopically complete resection of meningioma [91]. If a sub-total resection has been performed, the recurrence rate for WHO grade I meningiomas could outpass 50% within ten years [92]. The diagnostic accuracy of standard mpMRI is limited, especially in complex situations with bone infiltration or scar tissue. [^68^Ga]Ga-DOTA-SSTR PET has been proven to add incremental value to mpMRI in the case of TRC [79,82] (Figure 4). It is also useful to differentiate scar tissue from active tumors, with a high sensitivity of 90% [82]. It performs particularly better than mpMRI in the case of transosseous meningiomas, with a sensitivity of 97% and specificity of 100%, compared to 54% and 83%, respectively, for mpMRI [92].

### 5.5. [^177^Lu]Lu-DOTA-SSTR

The prognosis of patients with progressive meningioma after failure of surgery and radiotherapy is poor. Targeted radiation therapy with [^177^Lu]Lu-DOTA-SSTR, which is well recognized for the treatment of advanced neuroendocrine tumors, is increasingly used for patients with evolving and unresectable meningiomas. 

In a phase II clinical trial, Marincek et al. performed 74 treatment cycles on 34 patients, achieving disease stabilization among 23 of them (67%). Stable disease after treatment and high tumor uptake were associated with longer survival [93]. Another retrospective study evaluated the safety and efficacy of [^177^Lu]Lu-DOTA-SSTR in 20 patients with progressive treatment-refractory meningiomas. The treatment led to disease stabilization in 10 of 20 patients. Median progression-free survivals of 32.2 months for grade I tumors, 7.2 for grade II and 2.1 for grade III were reached. The median overall survival was 17.2 months in grade III patients and not reached for I and II at a median follow-up of 20 months [94]. Finally, in a study including seven patients with progressive intracranial meningioma a progression free survival at six months of 42.9% has been reported after a median of four cycles of ^177^Lu-DOTA-SSTR administered, [95].

[^68^Ga]Ga-DOTA-SSTR PET imaging seems essential before treatment to evaluate the tumor SSTR expression level for individualized treatment optimization.

## 6. PCNSL

PCNSL are non-Hodgkin lymphoma, most frequently represented by diffuse large B-cell lymphoma (DLBCL), confined to the brain, spinal cord, eyes and leptomeninges. Most PCNSL show a very intense [^18^F]FDG uptake. It can be used for differential diagnosis in both for immunocompetent and immunocompromised patients. In immunocompromised patients, [^18^F]FDG PET distinguishes correctly PCNSL from other brain infection with a sensibility of 100% and a specificity yielded from 75% to 100% [96]. In immunocompetent patients, [^18^F]FDG in addition to perfusion MRI enables to differentiate PCNSL from glioblastoma with very good diagnostic accuracy, reaching a sensitivity of 95% and a specificity of 96.4% [97]. The uptake intensity of PSNCL is usually more than twice that of brain physiological uptake [97] (Figure 5).

Concomitant systemic involvement impacts patients’ management. [^18^F]FDG PET has a higher diagnostic yield than computed tomography to detect extracranial lymphoma location and is the imaging method of reference for the staging of systemic DLBCL. Bertaux et al. reported that [^18^F]FDG PET revealed concomitant occult systemic lymphoma involvement in 8% of 130 PCNSL patients and was more sensitive than a combination of contrast-enhanced CT and bone-marrow biopsy [98]. In a very recent systematic review and meta-analysis, Park et al. stated that whole-body [^18^F]FDG PET should be preferred over CT in the initial workup of patients with suspected primary CNS lymphoma to detect occult systemic involvement [99]. 

## 7. Other Aspects of PET Imaging

### 7.1. “Unconventional” Tracers

Numerous innovative PET tracers have been (and are still continuously) developed that target several biological aspects of brain tumors and their micro-environment, as follows: blood flow, angiogenesis, hypoxia, neuroinflammation, mitotic activity and receptor binding [100]. Notably, perfusion can be assessed by [^15^O]H_2_O and [^13^N]NH_3_; neoangiogenesis can be highlighted by Arg-Gly-Asp peptide (RGD)-based PET tracers, especially [^18^F]FPPRGD2, [^64^Cu]DOTA-VEGF121 and [^89^Zr]Bevacizumab; neuroinflammation can be explored by multiple tracers targeting translocator protein (TSPO), mitotic activity can be revealed by [^18^F]Fluorothymidine (FLT); and several tracers specifically receptors such as C-X-C chemokine receptor type 4 (CXCR4), epidermal growth factor receptor (EGFR), transforming growth factor-β (TGF-β), fibroblast activation protein (FAP) and pyruvate kinase M2 (PKM2) [100].

In clinical practice, [^18^F]Fluorocholine (FCH) PET can be useful on a case-by-case for tumor characterization and differential diagnosis [81], even though it does not cross the intact blood–brain barrier. Alongi et al. reported a case in which [^18^F]FCH PET/CT had successively made differential diagnosis between cystic glioblastoma and intraparenchymal hemorrhage, supporting the potential use of this imaging biomarker in surgical or radio-surgical approach [101].

### 7.2. PET/MRI

Integrated PET/MRI systems have been increasingly used in expert centers in recent years. Such systems can provide a complete quick and accurate tumor evaluation by the synchronous acquisition of complementary modalities and thus reduce transportations of patients and increase their comfort. Nevertheless, since these high-end systems remain very expensive, and brain images can be perfectly fused after being acquired separately, the highest added value of integrated PET/MRI systems is yet limited to clinical research for many brain tumor types [102,103,104]. 

### 7.3. Additional Approaches in Image Analysis

Artificial intelligence (AI) has already become a reality in radiology. Since September 2020, the U.S. Centers for Medicare and Medicaid Services officially granted their first reimbursement of a radiology AI algorithm. We can reasonably expect a broader coverage of imaging AI software in clinical practice, including in nuclear medicine. In neuro-oncology, several studies concerning machine learning algorithms and radiomics have already been published, with encouraging results. For example, Russo et al. have developed a predictive model using machine learning based on radiomics, for discriminating between low-grade and high-grade CNS tumors in 56 patients who underwent [^11^C]MET PET [105]. Kebir et al. highlighted the potential use of radiomics on [^18^F]FET PET imaging to differentiate multiple sclerosis from glioma, and they concluded that machine learning can enhance lesions classification [106]. Qian et al. used radiomics features extracted from [^18^F]FDOPA PET to predict methylation of the O6-methylguanine methyltransferase (MGMT) gene promoter status in gliomas, with reasonable accuracy (nearly 80%) [107]. 

## 8. Conclusions

Molecular imaging with PET, can be of great value in the clinical management of primary and secondary brain tumors, especially as precision and personalized medicine continues to develop. The existing literature provides strong evidence that PET can efficiently supplement MRI in specific settings such as distinguishing recurrence from TRC in glioma and brain metastases. Beyond amino acid PET tracers for glioma, [^68^Ga]Ga-DOTA-SSTR PET and [^18^F]-FDG PET can play an essential role in the workup of meningiomas, metastases and PCNSL. These diagnostic procedures directly lead to benefits for patients suffering from brain tumors and could be more widely used in clinical routine as the availability of both tracers and imaging systems improves. When appropriately combined with mpMRI, PET imaging has been shown to be of incremental value at many time points in the course of several brain tumors, covering almost the whole diagnostic range in clinical neuro-oncology. Studies are still needed to strengthen the evidence level, specify its exact role in the different scenarios of clinical routine, homogenize practices and provide the community with clear guidelines to systematically implement PET imaging in neuro-oncology.

## Figures and Tables

**Figure 1 cancers-14-00879-f001:**
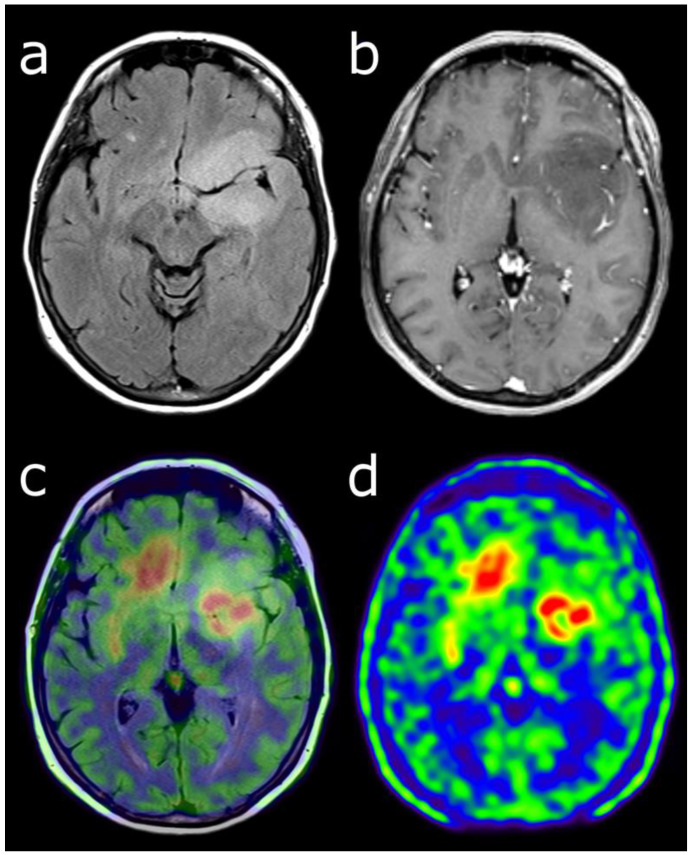
Characterization of a left temporo-fronto-insular brain lesion in a 60-year-old woman. This lesion was considered suggestive of grade II or III glioma according T2-FLAIR (**a**) and post-enhancement T1-weighted MRI (**b**). [18F]FDOPA PET images (fused with T2-FLAIR MRI (**c**) and PET only (**d**)) showed an uptake more intense than twice the normal cortex, predictive of high-grade tumor. Moreover, the lesion visualized with PET was much more extended than with MRI, particularly in the contralateral frontal lobe. Pathological analysis of biopsy samples revealed a WHO grade IV glioblastoma.

**Figure 2 cancers-14-00879-f002:**
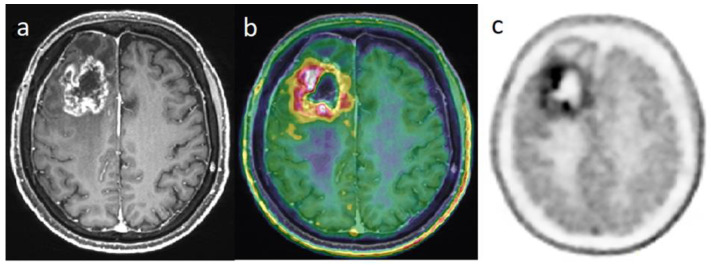
Differential diagnosis between recurrence versus radiation necrosis in a 70-year-old man who suffered from a right frontal glioblastoma IDH1-wt and was treated with surgery then chemoradiation with Temodal. Nine months later, there is an increase in edema in the site of the initial tumor. Post-enhancement T1-weighted MRI (**a**) revealed an increasing contrast enhancement which could correspond to a recurrence or a radionecrosis. [^18^F]FDOPA PET images (fused with T1-weighted MRI (**b**) and PET only (**c**)) showed an intense uptake of this lesion, with tumor/striata ratio = 1.5, suggesting tumor recurrence.

**Figure 3 cancers-14-00879-f003:**
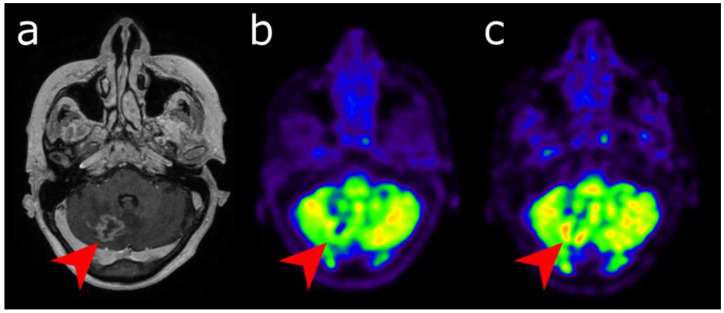
Differential diagnosis between recurrence versus radionecrosis in a 73-year-old woman who was treated with stereotactic radiation therapy 2 years earlier for a cerebellar metastasis (red arrow) of breast cancer. Post-enhancement T1-weighted MRI (**a**) revealed an increasing contrast enhancement. While standard [18F]FDG PET imaging performed 60 min (**b**) displayed no significant uptake, delayed images performed 4 h post-injection (**c**) revealed uptake higher than the background activity, suggesting tumor recurrence.

**Figure 4 cancers-14-00879-f004:**
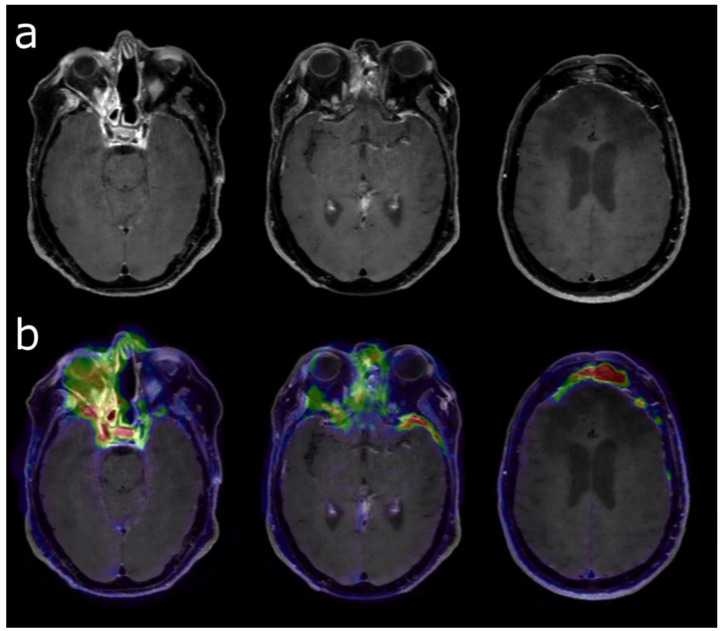
Post-enhancement T1-weighted MRI (**a**) before radiation therapy of a grade 2 skull base meningioma in a 49-year-old woman who was previously treated several times by surgery. [68Ga]Ga-DOTATOC PET (**b**) was performed to differentiate viable tumor (high uptake) vs. post-therapeutic changes (moderate uptake). Meningioma extension (to frontal and left temporal areas) was much more obvious using [68Ga]Ga-DOTATOC PET in addition to MRI.

**Figure 5 cancers-14-00879-f005:**
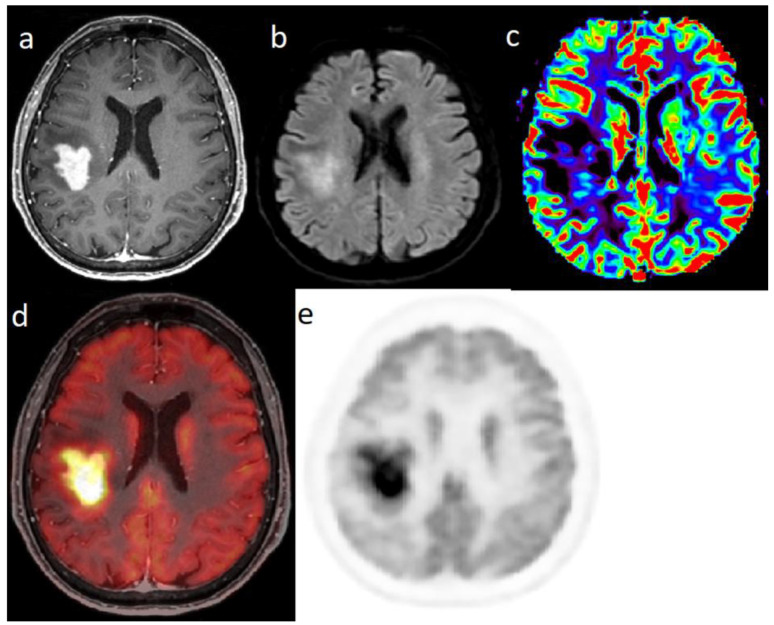
Characterization of a single right parietal brain lesion in an 81-year-old woman. Post-enhancement T1-weighted MRI (**a**) showed an intense increasing contrast enhancement; diffusion weighted imaging (**b**) showed a hypersignal and the there was no hyperperfusion in perfusion-weighted imaging (**c**). [18F]FDG PET (fused with T1-weighted MRI (**d**) and PET only (**e**)) showed a highly intense uptake: SUVmax = 37. Pathological analysis of biopsy samples confirmed a PCNSL (type DLBCL).

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
