# Peer review of "What Does PET Imaging Bring to Neuro-Oncology in 2022? A Review"

_cancers, 2022, doi:10.3390/cancers14040879_

Round 1

Reviewer 1 Report

The authors present the role of PET for primary and secondary brain tumors diagnosis, assessment, and follow-up. Despite that during the last years, PET is going to acquire an always increasing role in the management of such tumors, coupled with the MRI, and the literature shows increasing manuscript about this, the current manuscript provides relevant insides for clinical management. Whereas in form of narrative review (and not as a stronger structure as systematic review), I think that the present article could be helpful for clinicians. 

I only suggest adding, in a new paragraph, a brief overview of the role of "unconventional" tracers or additional approaches as machine learning (see i.e. Russo G et al,  Feasibility on the Use of Radiomics Features of 11[C]-MET PET/CT in Central Nervous System Tumours: Preliminary Results on Potential Grading Discrimination Using a Machine Learning Model. Curr Oncol. 2021 Dec 12;28(6):5318-5331. doi: 10.3390/curroncol28060444 AND Laudicella R et al, Unconventional non-amino acidic PET radiotracers for molecular imaging in gliomas. Eur J Nucl Med Mol Imaging. 2021 Nov;48(12):3925-3939. doi: 10.1007/s00259-021-05352-w. AND Alongi P et al. Choline-PET/CT in the Differential Diagnosis Between Cystic Glioblastoma and Intraparenchymal Hemorrhage. Curr Radiopharm. 2019;12(1):88-92. doi: 10.2174/1874471011666180817122427. PMID: 30117406.)

Minor issue: authors should avoid repeating some sentences within the same section (see " In this setting" in paragraph 3.1)

Author Response

We sincerely thank you for these kind and valuable comments which were of great help in revising this paper. Our responses follow. 

- I only suggest adding, in a new paragraph, a brief overview of the role of "unconventional" tracers or additional approaches as machine learning (see i.e. Russo G et al,  Feasibility on the Use of Radiomics Features of 11[C]-MET PET/CT in Central Nervous System Tumours: Preliminary Results on Potential Grading Discrimination Using a Machine Learning Model. Curr Oncol. 2021 Dec 12;28(6):5318-5331. doi: 10.3390/curroncol28060444 AND Laudicella R et al, Unconventional non-amino acidic PET radiotracers for molecular imaging in gliomas. Eur J Nucl Med Mol Imaging. 2021 Nov;48(12):3925-3939. doi: 10.1007/s00259-021-05352-w. AND Alongi P et al. Choline-PET/CT in the Differential Diagnosis Between Cystic Glioblastoma and Intraparenchymal Hemorrhage. Curr Radiopharm. 2019;12(1):88-92. doi: 10.2174/1874471011666180817122427. PMID: 30117406.)

R: We have added the 7th chapter entitled "Other aspects of PET imaging", in which we have discussed these intereting topics on "unconventional" tracers and additional approaches as machine learning. The corresponding citations have been made.

- Minor issue: authors should avoid repeating some sentences within the same section (see " In this setting" in paragraph 3.1)

R: We have delected most of them and have an English-editing service. Thank you for your advices.

Reviewer 2 Report

This paper compreshensively reviewed application of PET in brain tumors, highligted the values of PET in detection and differential diagnosis. The draft was well organized and well written. 

I have a few of minor concerns.

  1. Expectation. should give descripton on future development of new technique on brain tumors.
  2.  PET/MRI, suggst adding a few of sentences on its values .

Author Response

We sincerely thank you for your constructive criticism and valuable comments

which were of great help in revising this paper. Our responses follow. 

1.Expectation. should give descripton on future development of new technique on brain tumors

R: We have added a new paragraph entitled  "7. Other aspects of PET imaging" in which we discussed new approches as radiomics analysis and machine learning. 

2. PET/MRI, suggst adding a few of sentences on its values 

R: Indeed, it's a relevant suggestion and we have added several sentences on this topic, in 7.2.

Reviewer 3 Report

Interesting review on PET applications in neuroimaging. The topic discussion is complete and well balanced. No major weakness

Author Response

We sincerely thank you for your valuable support and we are very pleased that our manuscript meet your requirements.

We have solicited an English language editing services for a better presentation.

Round 2

Reviewer 1 Report

I think that the manuscript is now suitable for publication in the present frm